# Impact of 1-Hour Bundle Achievement in Septic Shock

**DOI:** 10.3390/jcm10030527

**Published:** 2021-02-02

**Authors:** Byuk Sung Ko, Sung-Hyuk Choi, Tae Gun Shin, Kyuseok Kim, You Hwan Jo, Seung Mok Ryoo, Yoo Seok Park, Woon Yong Kwon, Han Sung Choi, Sung Phil Chung, Gil Joon Suh, Hyunggoo Kang, Tae Ho Lim, Donghee Son, Won Young Kim

**Affiliations:** 1Department of Emergency Medicine, College of Medicine, Hanyang University, Seoul 04763, Korea; postwinston@gmail.com (B.S.K.); emer0905@gmail.com (H.K.); erthim@gmail.com (T.H.L.); 2Department of Emergency Medicine, Korea University Guro Hospital, Seoul 08308, Korea; kuedchoi@korea.ac.kr; 3Department of Emergency Medicine, Samsung Medical Center, Sungkyunkwan University School of Medicine, 81 Irwon-ro, Gangnam-gu, Seoul 06351, Korea; tackles@naver.com; 4Department of Emergency Medicine, CHA University School of Medicine, CHA Bundang Medical Center, Seongnam-si 13449, Gyeonggi-do, Korea; dreinstein70@gmail.com; 5Department of Emergency Medicine, Seoul National University Bundang Hospital, Seongnam-si 13620, Gyeonggi-do, Korea; drakejo@snubh.org; 6Department of Emergency Medicine, College of Medicine, Asan Medical Center, University of Ulsan, Seoul 05505, Korea; chrisryoo@gmail.com; 7Department of Emergency Medicine, Yonsei University College of Medicine, Seoul 03722, Korea; pys0905@yuhs.ac; 8Department of Emergency Medicine, Seoul National University Hospital, Seoul 03080, Korea; kwy711@hanmail.net (W.Y.K.); suhgil@snu.ac.kr (G.J.S.); 9Department of Emergency Medicine, Kyung Hee University Hospital, Seoul 02453, Korea; hsg3748@hanmail.net; 10Department of Emergency Medicine, Gangnam Severance Hospital, Seoul 06273, Korea; emstar@naver.com; 11Biostatistical Consulting and Research Lab, Medical Research Collaborating Center, Hanyang University, Seoul 04763, Korea; sdh9123@hanyang.ac.kr

**Keywords:** sepsis, septic shock, 1-h bundle, outcome, mortality, emergency department

## Abstract

This study aimed to address the impact of 1-hr bundle achievement on outcomes in septic shock patients. Secondary analysis of multicenter prospectively collected data on septic shock patients who had undergone protocolized resuscitation bundle therapy at emergency departments was conducted. In-hospital mortality according to 1-h bundle achievement was compared using multivariable logistic regression analysis. Patients were also divided into 3 groups according to the time of bundle achievement and outcomes were compared to examine the difference in outcome for each group over time: group 1 (≤1 h reference), group 2 (1–3 h) and group 3 (3–6 h). In total, 1612 patients with septic shock were included. The 1-h bundle was achieved in 461 (28.6%) patients. The group that achieved the 1-h bundle did not show a significant difference in in-hospital mortality compared to the group that did not achieve the 1-h bundle on multivariable logistic regression analysis (<1 vs. >1 h) (odds ratio = 0.74, *p* = 0.091). However, 3- and 6- h bundle achievements showed significantly lower odds ratios of in-hospital mortality compared to the group that did not achieve the bundle (<3 vs. >3 h, <6 vs. >6 h, odds ratio = 0.604 and 0.458, respectively). There was no significant difference in in-hospital mortality over time for group 2 and 3 compared to that of group 1. One-hour bundle achievement was not associated with improved outcomes in septic shock patients. These data suggest that further investigation into the clinical implications of 1-h bundle achievement in patients with septic shock is warranted.

## 1. Introduction

An estimated 48.9 million cases of sepsis have been reported, accounting for 19.7% of all global deaths [1,2]. The incidence of sepsis increases with advanced age, comorbidities, and immunocompromised status [3,4]. This may also be due to increased detection of early sepsis as a result of intensive sepsis education and awareness campaigns. Despite advances in recent critical care, sepsis remains a serious disease with high mortality and morbidity.

Early identification and management is essential in septic patients. In 2005, the Surviving Sepsis Campaign (SSC) suggested a protocolized bundle therapy to facilitate implementation at the bedside with a defined target [5,6]. SSC bundles were revised from 6-h to 3-h bundles in 2015. The guidelines published in 2018 state that this resuscitation bundle treatment should be initiated within 1 h of the emergency department (ED) triage time or the earliest chart annotation if presenting from another care venue, named the 1-h bundle [7]. The 1-h bundle is composed of the following five elements: measuring the lactate level, obtaining blood culture prior to administration of antibiotics, administering broad-spectrum antibiotics, beginning rapid administration of 30 mL/kg crystalloid fluid for hypotension or lactate ≥4 mmol/L, and administering vasopressors if the patient is hypotensive during or after fluid resuscitation to maintain mean arterial pressure (MAP) at ≥65 mmHg within 1 h from sepsis recognition. However, in October 2019, the definition of time zero changed to the time of shock recognition [8]. This recommendation is based on a study that reported a significant reduction in mortality associated with timely completion of 3-h bundles [9]. Although this study did not address the direct effect of the 1-h bundle, it provided supporting evidence. However, there are concerns regarding the lack of evidence supporting the cut-off of 1-h and a concern about the 1-h bundle causing hasty management decisions, inappropriate fluid administration, and indiscriminate use of broad-spectrum antibiotics [10,11]. The quality of evidence supporting the individual elements of the bundle varies from low to moderate [7]. To the best of our knowledge, no study has examined the prognostic value of the 1-h bundle achievement in sepsis and septic shock.

We aimed to evaluate the impact of 1-h bundle achievement in patients with septic shock visiting the ED. We hypothesized that 1-h bundle achievement is not associated with significantly better outcomes than 3- or 6-h bundle achievement.

## 2. Methods

### 2.1. Study Design

Secondary analysis of prospectively collected data was conducted involving 10 Korean university-affiliated hospital EDs using data from the Korean Shock Society septic shock registry from October 2015 to December 2018. Patients aged ≥19 years who met the inclusion criteria (evidence of refractory hypotension or hypoperfusion in patients with suspected or confirmed infection) were included [12,13]. Hypotension was defined as systolic blood pressure (SBP) <90 mmHg, MAP <70 mmHg, or an SBP decrease of >40 mmHg. Refractory hypotension was defined as persistent hypotension despite the administration of fluid challenge (30 mL/kg of crystalloid fluid) or the requirement of vasopressors to maintain an SBP of ≥90 mmHg or a MAP of ≥70 mmHg. Hypoperfusion was defined as the presence of serum lactate levels ≥4 mmol/L. Patients were excluded if they had given “do not attempt resuscitation” orders, met the inclusion criteria 6 h after arrival at the ED, were transferred from other hospitals and did not meet the inclusion criteria on arrival at the ED, or were transferred directly from the ED to other hospitals. The institutional review board of each institution approved the study protocol, and informed consent was obtained from all patients before data collection. In this registry, information regarding the time of lactate measurement, blood culture, antibiotic administration, fluid administration, and use of vasopressors was recorded. A detailed description of the registry has been presented elsewhere [14,15,16].

In addition to the general septic shock registry described above, this study used an additional design. In our registry, overall cohorts were composed of patients who visited the ED directly or were transferred from another hospital. The cohort was limited to patients presenting directly to the EDs of the study institutions. We also only included patients who were enrolled for refractory hypotension because data regarding infusion of 30 mL/kg of crystalloid fluid of patients enrolled for hypoperfusion were not investigated in our registry. Patients with missing information on any bundle component were excluded. Patients with missing information on their outcomes were also excluded.

### 2.2. Definitions and Outcomes

Patients were divided into 3 groups according to the interval from recognition of sepsis and septic shock to bundle achievement: group 1 (≤1 h; reference), 2 (1–3 h) and 3 (3–6 h). We defined the recognition of septic shock as the time of refractory hypotension (persistent hypotension despite administration of fluid challenge).

If any of the five elements was not achieved within the specified time, bundle achievement was defined as failed. The outcomes of all groups were compared to those of the reference group. The decision to perform each element of the bundle was made by the treating physician. However, all the participating hospitals adhered to the recommendations of the SSC guidelines. The primary outcome of this study was in-hospital mortality; the secondary outcomes were 28-day and 90-day mortality.

### 2.3. Statistical Analyses

Continuous variables were analysed as means ± standard deviation or medians with interquartile ranges, as appropriate, and categorical variables were analysed as absolute or relative frequencies. Continuous variables are presented by median (Q1–Q3) and tested by the Kruskal–Wallis test. The chi-squared test or Fisher’s exact test were used for categorical variables.

The univariate and multivariable logistic regression analyses of each hour bundle achievement for predicting outcomes were conducted (<1-h vs. >1-h, <3-h vs. >3-h, and <6-h vs. >6-h, 2-group comparison). Multivariable logistic regression analysis was used to assess each bundle achievement group based on outcomes, with adjustment for confounding variables that were significant on univariate analysis. Variables yielding *p* < 0.2 on univariate analysis were entered in a backward fashion in the multivariable analysis.

Patients were also divided into 3 groups according to the interval to bundle achievement, and outcomes were compared to examine the linear relationship in outcome for each group over time: group 1 (≤1 h; reference), 2 (1–3 h), and 3 (3–6 h). Comparisons of 2 groups with the reference group were analysed using a multivariable logistic model.

A two-sided *p*-value <0.05 was considered statistically significant, and the Bonferroni-corrected threshold for statistical significance was computed and applied in each category. All statistical analyses were performed using SAS version 9.4 (SAS Institute; Cary, NC, USA) and R version 3.5.2 (R Foundation for Statistical Computing, Vienna, Austria).

## 3. Results

### 3.1. Participant Characteristics

A total of 1777 patients with refractory hypotension visited the ED directly during the study period. Of these, 165 patients with missing information on bundle achievement were excluded (Figure 1). Finally, 1612 patients were included. The number of patients in group 1, 2 and 3 were 461, 637 and 293, respectively. The baseline characteristics of the patients in group 1, 2 and 3 are shown in Table 1. The mean patient age of group 1, 2 and 3 were 68, 70 and 71 years, respectively. There was no significant difference in regard to age and male proportion. In-hospital mortality of group 1 was 13.8% (*n* = 64). The median time from ED visit to shock recognition was 87 min (interquartile range, 26–150 min). The 1-h bundle was achieved in 461 patients (28.6%). The mean initial SBP was not significantly different between the group 1, 2 and 3 (*p* = 0.152). The median lactate level between the group 1, 2 and 3 was significantly different (2.5 vs. 2.7 vs. 3.2 mmol/L, *p* < 0.001). In-hospital mortality was not significantly different between 3 groups (13.8% vs. 16.9% vs. 20.1%, *p* = 0.075). Univariate logistic regression analysis of 1-, 3- and 6-h bundles (2 group comparison) was performed to predict in-hospital mortality (Table 2). Achievement of a 1-h bundle was significantly associated with lower in-hospital mortality (*p* = 0.005). The odds ratios (ORs) of 3- and 6-h bundle achievements for in-hospital mortality were significantly low (OR = 0.603 and 0.511, respectively; *p* < 0.001 both). Old age, Sequential Organ Failure Assessment score, and Acute Physiologic Assessment and Chonic Health Evaluation score were associated with higher in-hospital mortality in the univariate analysis (Appendix A).

### 3.2. Multivariate Logistic Regression Analysis of 1-, 3- and 6- h Bundle Achievement to Predict in-Hospital Mortality (2 Group Comparison)

The adjusted OR of 1-h bundle achievement (<1 vs. >1 h, 2-group comparison) was 0.74 in predicting in-hospital mortality on multivariable logistic regression, but it was not significant (confidence interval [CI]: 0.522–1.049, *p* = 0.091) (Table 2). Three-hour bundle achievement (<3 vs. >3 h, 2-group comparison) was significantly associated with lower in-hospital mortality (OR = 0.604, CI: 0.446–0.819, *p* = 0.001); 6-h bundle achievements were also associated with lower in-hospital mortality (OR = 0.458, CI: 0.312–0.672, *p* <0.01).

### 3.3. Comparison of Outcomes to Examine the Linear Relationship for 3 Groups over Time: Group 1 (≤1 h; Reference), 2 (1–3 h), and 3 (3–6 h)

Multivariable analyses were performed to examine the effect of time delay in bundle achievement, with the 1-h bundle achievement group as the reference group. There were 461 patients in group 1 (reference group), and were 637 and 293 patients in groups 2 and 3, respectively. There was no significant difference or trend over time in in-hospital mortality of group 2 and 3 compared to group 1 on multivariable logistic regression analysis (Appendix A).

Multivariable analyses were conducted to predict the 28-day mortality of group 2 and 3, with the group 1 as the reference group. There was no significant difference in the 28-day mortality between the 2, 3 groups and the group 1 on multivariable logistic regression analysis (Appendix A). There was no significant difference in the 90-day mortality between the groups (Appendix A).

## 4. Discussion

In this study, we found that the achievement of a 1-h bundle was not independently associated with improved outcomes in patients with septic shock in the ED. No linear association was observed between the time delay in bundle achievement and outcomes. However, 3- and 6-h bundle achievement were all associated with a better outcome.

To the best of our knowledge, this is the first study to address the impact of 1-h bundle achievement on the outcomes of patients with septic shock. Several studies have addressed individual elements of bundle treatment in sepsis or septic shock, but no study has investigated the impact of whole bundle achievement. Our study population was selected from a prospective, multicenter study with a large sample size. Our registry had all information regarding bundle elements, especially time variables, although it was not constructed to examine the impact of 1-h bundle achievement. There are controversies regarding the 1-h bundle treatment, and this study may be the basis for future research and may help clinicians in sepsis care.

The effectiveness of bundle therapy in sepsis and septic shock treatment is controversial [11,17,18,19]. In a multicenter retrospective cohort study, the group that failed to achieve severe sepsis and septic shock performance measure bundle (SEP-1) had a higher crude mortality than the group that achieved well, but no significant difference was found after adjusting for clinical variables and disease severity [20]. There were five elements in SEP-1, similar to those in the 1-h bundle, but the time limits were 3 h and 6 h. In a study by Baghdadi et al., timely lactate measurement was associated with reduced mortality, but full SEP-1 adherence was not associated with improved outcome in patients with hospital-onset or community-onset sepsis [21]. According to one systematic review, no high- or moderate-level evidence showed that SEP-1 or its haemodynamic interventions improve the survival rate of adults with sepsis [22]. In a study analysing the effect of the 3-h bundle of sepsis care on 49,311 patients in 149 hospitals in New York, rapid bundle completion was associated with reduced in-hospital mortality [9]. A longer time to bundle completion was associated with higher rates of risk-adjusted in-hospital mortality. However, as the time to achieve bundle treatment was delayed, outcomes were not significantly different in our study despite the bundle achievement group showing significantly better outcomes than the failed group when comparing 2 groups (that is, 2-h achievement vs. 2-h failure). Our study consisted of five bundle treatment elements and the study in New York consisted of three elements (blood culture, lactate level measurement, and antibiotic administration); hence, comparison would be difficult.

Our study has several limitations. First, our data were not constructed to investigate 1-h bundle treatment. Our data were not part of a performance improvement initiative, as we performed secondary analysis of prospectively collected data. Performance improvement examining a 1-h bundle would be better to validate its prognostic value. Second, patients with hypoperfusion were not included in this study. Hence, these data are generalizable to the overtly hypotensive population with sepsis and not the cryptic shock population. Third, it is unclear whether there was a start of rapid administration of 30 mL/kg crystalloid fluid within 1 h of shock recognition because we defined the time of shock recognition as the time of refractory hypotension despite fluid administration. It could be argued that our analysis did not consist of five bundle treatments and that it consisted of only four bundle treatments. However, it is likely that start of rapid fluid administration might be achieved within 1-h because the median time from ED triage to fluid completion was 87 min. In the SSC guidelines, it is mandated that rapid fluid administration should be initiated within 1-h of shock recognition. We believe that start of fluid administration was performed as rapid as fast given median time. Fourth, the inherent limitations of a registry-based observational study should be acknowledged. Although our registry collected all variables related to bundle treatment elements, we cannot exclude the influence of unmeasured confounders that may have affected the results despite adjustment for the differences in the baseline risk factors with multivariable analysis. Finally, the differences in the outcomes of each participating hospital were not addressed.

## 5. Conclusions

The 1-h bundle was achieved by 28.6% of septic shock patients who were treated with a protocolized bundle therapy in the ED and 1-h bundle was not independently associated with better outcomes. No linear association was observed between the time delay in bundle achievement and patient outcome. However, 3- and 6-h bundle achievement were all associated with better outcomes compared to those who failed in comparison between the 2 groups. These data suggest that further investigation into the clinical implications of 1-h bundle achievement in patients with septic shock is warranted.

## Figures and Tables

**Figure 1 jcm-10-00527-f001:**
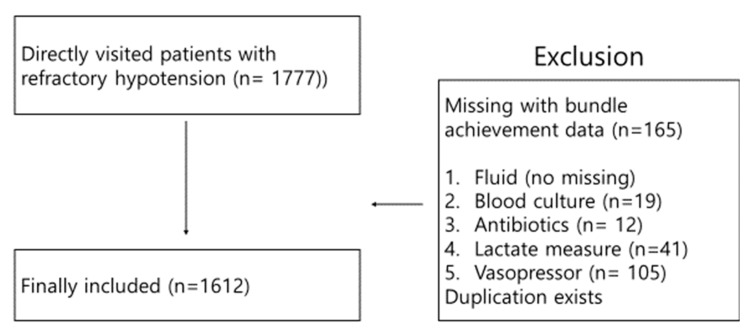
Patient selection flow diagram.

**Table 1 jcm-10-00527-t001:** Comparison of and clinical characteristics of the 3 bundle groups.

Variables	Group 1 (<1 h) (*n* = 461)	Group 2 (1–3 h) (*n* = 637)	Group 3 (3–6 h) (*n* = 293)	*p* Value
Age, years	68 (60–76)	70 (59–78)	71 (60–78)	0.063
Male, *n* (%)	255 (55.3)	352 (55.2)	173 (59)	0.514
Initial vital signs				
SBP, mm Hg	91 (77–111)	89 (74–108)	88 (74–107)	0.152
DBP, mm Hg	56 (48–67)	54 (45–64)	54 (46–64)	0.057
Heart rate, beat per min	111 (94–128)	111 (93–127)	111 (94–130)	0.899
Respiratory rate, per min	20 (18–24)	20 (18–22)	20 (18–23)	0.054
Body temperature, ℃	38.1 (37.1–38.9)	38 (36.9–39)	38 (36.8–38.8)	0.256
Comorbidities, *n* (%)				
Hypertension	184 (39.9)	252 (39.5)	103 (35.1)	0.361
Diabetes mellitus	105 (22.7)	180 (28.2)	73 (24.9)	0.114
Cardiac disease	81 (17.5)	72 (11.3)	33 (11.2)	0.005
COPD	40 (8.6)	46 (7.2)	20 (6.8)	0.566
CKD	33 (7.1)	47 (7.3)	17 (5.8)	0.668
Chronic liver disease	57 (12.3)	68 (10.6)	25 (8.5)	0.253
Infection site, *n* (%)				
Respiratory	123 (26.6)	128 (20.1)	66 (22.5)	0.036
Urinary tract	93 (20.1)	140 (21.9)	51 (17.4)	0.271
Gastrointestinal tract	48 (10.4)	104 (16.3)	37 (12.6)	0.016
Hepato-biliary and pancreas	85 (18.4)	96 (15.1)	57 (19.4)	0.167
Others *	20 (4.3)	32 (5)	18 (6.1)	0.543
Lactate, mmol/L	2.5 (1.7–3.5)	2.7 (1.6–4.8)	3.2 (1.9–5)	<.0001
SOFA	6 (4–8)	6 (4–8)	6 (4–8)	0.635
APACHE 2	19 (13–25)	19 (13–25)	21 (15–27)	0.021
Positive blood culture, *n* (%)	208 (45.1)	289 (45.3)	135 (46)	0.966
Outcomes, *n* (%)				
In-hospital mortality	64 (13.8)	108 (16.9)	59 (20.1)	0.075
28-day mortality (*n* = 1292)	57 (13.2)	98 (16.4)	55 (20.6)	0.037
90-day mortality (*n* = 1148)	102 (26.1)	138 (26.4)	75 (31.7)	0.243

APACHE 2: Acute Physiology and Chronic Health Evaluation 2, CKD: chronic kidney disease, COPD: chronic obstructive pulmonary disease, DBP: diastolic blood pressure, SOFA: Sequential Organ Failure Assessment. Continuous variables are presented by median (Q1–Q3) and tested by the Kruskal–Wallis test, and categorical variables are presented by *n* (%) and tested by the chi-squared test. * Others includes soft tissue, central nervous system, catheter related, blood stream and endocarditis as infection site.

**Table 2 jcm-10-00527-t002:** Univariate and multivariable logistic regression analysis for in-hospital mortality.

Variables	Unadjusted OR	95% CI of OR	*p* Value	Adjusted OR	95% CI of OR	*p* Value
Bundle achievement (2 group comparison)						
1-h bundle (<1 h vs. >1 h)	0.649	0.481–0.877	0.005	0.74	0.522–1.049	0.091
3-h bundle (<3 h vs. >3 h)	0.603	0.465–0.783	<0.001	0.604	0.446–0.819	0.001
6-h bundle (<6 h vs. >6 h)	0.511	0.369–0.707	<0.001	0.458	0.312–0.672	<0.01

CI: confidence interval, OR: odds ratio.

## Data Availability

Data available on request due to restrictions eg privacy or ethical.

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
