# Peer review of "Impact of 1-Hour Bundle Achievement in Septic Shock"

_jcm, 2021, doi:10.3390/jcm10030527_

Round 1
Reviewer 1 Report
Review of JCM submission, 1069622, “Impact of 1-Hour Bundle Achievement in Septic Shock.”
Major Comments:
Specific Comments:
Abstract:
“The group achieving a 1-hour bundle did not show a significant difference in in-hospital mortality compared to the group that did not achieve the bundle (<1 vs. >1 hour) (odds ratio = 0.74, p = 0.091). However, groups 2, 3, 4, 5, and 6 showed significantly lower odds ratios of in-hospital mortality compared to the group that did not achieve the bundle (2-group comparison, odds ratio = 0.733, 0.604, 0.541, 0.532, and 0.458, respectively).”
This is confusing. “compared to the group that did not achieve the bundle (<1 vs. >1 hour)” à not achieving the bundle here appears to mean the “1 hour bundle”.
While: “However, groups 2, 3, 4, 5, and 6 showed significantly lower odds ratios of in-hospital mortality compared to the group that did not achieve the bundle.” à not achieving the bundle here appears to mean not achieving the sepsis bundle at all: at anytime in their care.
Please discuss and clarify.
Introduction:
6 hour bundle --> 3 hour bundle --> resuscitation bundle initiated within one hour in the Emergency Department. You need to put the components of the resuscitation bundle into the Introduction for context (I realize they are in your Methods). Some readers will not know the specific elements of the SSC bundle. Alternatively, could describe how the bundle addresses perfusion (lactate), source control (blood cultures and antibiotics), and blood pressure, and then say see Methods for individual elements. Please modify/add.
Methods:
“to avoid confusion.” This phrasing should be corrected. The reason for excluding transferred patients is that the primary outcome being studied (the impact of achieving the 1 hour bundle) is not tested in the same way because it is not performed at the study institutions; in addition, the quality of data for transfers, especially for time points, is usually less rigorous. Please change phrasing to state that the cohort was limited to patients presenting directly to the EDs of the study institutions.
“Patients were divided into six groups according to the interval from recognition of sepsis and septic shock to bundle achievement: group 1 (≤1 h; reference).” The SSC 1 hour bundle defines time zero as time of ED triage, the validity of which has been debated. Why are you defining “time zero” as time of “recognition of sepsis?” Please explain.
Also, “We defined the recognition of septic shock as the time of refractory hypotension (persistent hypotension despite administration of fluid challenge).” Please describe further. If someone presents to ED triage @ 13:00 and has SBP of 80mmHg and after 30cc/kg @ 13:45 they still have SBP of 82mmHg, was the time of refractory hypotension 13:00 or 13:45? Also, it seems problematic to define the time of starting the 1 hour bundle dependent on the completion of an element (fluid of the bundle). Isn’t this falsely inflating the length of the first hour and altering the analysis?
Results:
“A significantly lower proportion of in-hospital mortality was observed in the group that achieved a 1-hour bundle than the group that did not to achieve the bundle (13.8% vs. 19.9%, p = 0.004).” In the abstract, you state that there was no difference in mortality between the completed 1 hour bundle and didn’t complete 1 hour bundle groups. Which is it? I assume that these data are the univariate analysis. After reading multiple times, it is clear that the abstract is presenting the multivariable analysis results while this is presenting the univariate. I would present the Patient Characteristics, then the Univariate Analysis, then the Multivariable Logistic Regression and present clearly. Please clarify.
“The odds ratios (ORs) of 2-, 3-, 4-, 5-, and 6-hour bundle achievements for in-hospital mortality were significantly low (OR = 0.64, 0.603, 0.574, 0.569, and 0.511, respectively; p = 0.001, <0.001, <0.001, <0.001, and <0.001, respectively).” Compared to what as reference? Please clarify. This only becomes clear in the multivariable logistic regression analysis section where you let us know that you are comparing < 2 hour to > 2 hour; < 3 hour to > 3 hour, etc. Please correct so reads more clearly.
What are “others” that are comorbidities in Table 1 and used in the Multivariable Analysis? Please define others.
pp 7-8, multivariable analysis comparing 1 hour to the other time intervals for in-house, 28 day, and 90 day mortality. I find this very confusing. This relates to my concerns above about what is the comparison group. You haven’t articulated this clearly in Methods, that you are comparing <1 hour to > 1 hour, then < 2 hour to > 2 hour but also comparing < 1 hour as reference to > 1 hour, < 1 hour to > 2 hour, etc. If this is the approach you want to take, need to make much clearer. Also, please explain why (rationale for) you would do both?
Pp 8-9: “Patients in whom blood culture was conducted within 1 and 2 h did not show a significant difference in in-hospital mortality compared to those for whom it was not (p = 0.228 and 0.087, respectively). Patients who underwent blood culture within 3, 4, 5, and 6 h had significantly lower in-hospital mortality rates than those who did not (p = 0.003, 0.007, 0.016, and 0.005, respectively).” Is this univariate or multivariable analysis? Also, again, compared to what as reference? Please clarify.
Lactate within 5 hours, lower mortality. I find all of these time points for different variables very confusing and distracting from the central message of the manuscript. It feels like data dredging. If you explore enough variables, you will find significance. For these 1 hour bundle variables, you are exploring enough that you have to assume you will find something significant (1 in 20). I recommend narrowing down to the key variables you want to analyze.
Discussion:
“In a single-centre retrospective study, the group that failed to achieve severe sepsis and septic shock performance measure bundle (SEP-1) had a higher crude mortality than the group that achieved well, but no significant difference was found after adjusting for clinical variables and disease severity [20].” This statement mischaracterizes the Rhee study. It is not a single-centre study. It is a multicenter retrospective cohort study. Rhee, C.; Filbin, M.; Massaro, A.F.; Bulger, A.; McEachern, D.; Tobin, K.A.; Kitch, B.; Thurlo-Walsh, B.; Kadar, A.; Koffman, A. 
Compliance with the national SEP-1 quality measure and association with sepsis outcomes: a multicenter retrospective cohort study. Crit. Care Med. 2018, 46, 1585. Please correct.
Evans study. Please justify the validity of comparing your adult study to the results of a pediatric sepsis study. How can you compare these results? I would recommend deleting reference to the Evans’ study.
Limitations. There is no way to know that low compliance with the 1 hour bundle is because clinicians were unaware of the study. This may be the best compliance you can get due to limitations in meeting aspects of the bundle within 1 hour. I would delete this sentence.
“Third, it is unclear whether there was a start of rapid administration of 30 mL/kg crystalloid fluid within 1 hour of shock recognition because we defined the time of shock recognition as the time of refractory hypotension despite fluid administration. It could be argued that our analysis did not consists of five bundle treatments and that it consists of only four bundle treatment. However, it is likely that start of rapid fluid administration might be achieved within 1 h because the median time from ED triage to fluid completion was 87 min.” From my perspective, this is the biggest limitation of this study and its approach to data analysis. I derive a different conclusion from the definition of “the time of shock recognition as the time of refractory hypotension despite fluid administration.” From my perspective this means that your one hour window, as defined by SSC, is probably more likely a 2 hour window, etc. This needs to be clearly stated and I am not sure that you investigate a 1 hour window at all. Please comment and justify how you have actually addressed a 1 hour window.
Author Response
First of all, thank you very much for your interest in our paper and your excellent comments.
Please check the line by line responses to the reviewer’s comments and the results of re-analysis.
Reviewer 1
Abstract:
“The group achieving a 1-hour bundle did not show a significant difference in in-hospital mortality compared to the group that did not achieve the bundle (<1 vs. >1 hour) (odds ratio = 0.74, p = 0.091). However, groups 2, 3, 4, 5, and 6 showed significantly lower odds ratios of in-hospital mortality compared to the group that did not achieve the bundle (2-group comparison, odds ratio = 0.733, 0.604, 0.541, 0.532, and 0.458, respectively).”
This is confusing. “compared to the group that did not achieve the bundle (<1 vs. >1 hour)” à not achieving the bundle here appears to mean the “1 hour bundle”.
Response) Yes, reviewer’s point is correct. We compared the group that achieved 1-hr bundle with the group that did not achieve through multivariate analysis and found out that there was no significant difference between 2 groups. According to reviewer’s recommendations, the abstract has been revised as follows.
“The group that achieved the 1-hr bundle did not show a significant difference in in-hospital mortality compared to the group that did not achieve the 1-hr bundle on multivariable logistic regression analysis (<1 vs. >1 hr) (odds ratio = 0.74, p = 0.091)”
While: “However, groups 2, 3, 4, 5, and 6 showed significantly lower odds ratios of in-hospital mortality compared to the group that did not achieve the bundle.” à not achieving the bundle here appears to mean not achieving the sepsis bundle at all: at anytime in their care.
Please discuss and clarify.
Response) We defined failure to achieve bundle treatment if any of the five bundle components were not achieved within a specific time (<2 vs. >2 hr, <3 vs. >3 hr, and so on, 2-group comparison). The purpose of this paper is to analyze how the achievement of 1-hr bundle associate with outcomes in septic shock, and to analyze whether there is a difference in outcome of patients over time compared to the group achieving 1-hr bundle as reference. In addition, you and reviewer 2 gave comments that this paper is very complicated, so the difference of outcome between the group that achieves the bundle within 1-hr compared to the existing SSC guideline's standard of achieving the bundle within 3-hr and 6-hr. The difference between the achievement time delay and the prognosis was also analyzed by changing from 6 groups to 3 groups. Therefore, it has been decided to delete the issue that could be confusing (<2 vs. >2 hr and <4 vs. >4 hr and <5 vs. >5 hr). The abstract has been changed as follows, so please check.
“This study aimed to address the impact of 1-hr bundle achievement on outcomes in septic shock patients. Secondary analysis of multicenter prospectively collected data on septic shock patients who had undergone protocolized resuscitation bundle therapy at emergency departments was conducted. In-hospital mortality according to 1-hr bundle achievement were compared using multivariable logistic regression analysis. Patients were also divided into 3 groups according to the time of bundle achievement and outcomes were compared to examine the difference in outcome for each group over time: group 1 (≤1 hr; reference), 2 (1–3 hr) and 3 (3–6 hr). In total, 1,612 patients with septic shock were included. The 1-hr bundle was achieved in 461 (28.6%) patients. The group that achieved the 1-hr bundle did not show a significant difference in in-hospital mortality compared to the group that did not achieve the 1-hr bundle on multivariable logistic regression analysis (<1 vs. >1 hr) (odds ratio = 0.74, p = 0.091). However, 3- and 6- hr bundle achievements showed significantly lower odds ratios of in-hospital mortality compared to the group that did not achieve the bundle (<3 vs. >3 hr, <6 vs. >6 hr, odds ratio = 0.604 and 0.458, respectively). There was no significant difference in in-hospital mortality over time for group 2 and 3 compared to that of group 1. One-hour bundle achievement was not associated with improved outcomes in septic shock patients. These data suggest that further investigation into the clinical implications of 1-hr bundle achievement in patients with septic shock is warranted.”
Introduction:
6 hour bundle --> 3 hour bundle --> resuscitation bundle initiated within one hour in the Emergency Department. You need to put the components of the resuscitation bundle into the Introduction for context (I realize they are in your Methods). Some readers will not know the specific elements of the SSC bundle. Alternatively, could describe how the bundle addresses perfusion (lactate), source control (blood cultures and antibiotics), and blood pressure, and then say see Methods for individual elements. Please modify/add.
Response) Thank you for the valuable comments and recommendations. According to reviewer’s suggestion, the description of the 1-hr bundle elements have been inserted into the introduction as follows.
The 1-hr bundle is composed of the following five elements: measuring the lactate level, obtaining blood culture prior to administration of antibiotics, administering broad-spectrum antibiotics, beginning rapid administration of 30 mL/kg crystalloid fluid for hypotension or lactate ≥4 mmol/L, and administering vasopressors if the patient is hypotensive during or after fluid resuscitation to maintain MAP at ≥65 mmHg within 1 h from sepsis recognition.
Methods:
“to avoid confusion.” This phrasing should be corrected. The reason for excluding transferred patients is that the primary outcome being studied (the impact of achieving the 1 hour bundle) is not tested in the same way because it is not performed at the study institutions; in addition, the quality of data for transfers, especially for time points, is usually less rigorous. Please change phrasing to state that the cohort was limited to patients presenting directly to the EDs of the study institutions.
Response) We totally agreed with reviewer’s opinion. According to reviewer’s suggestion, the method section has been changed as follows.
“The cohort was limited to patients presenting directly to the EDs of the study institutions”
“Patients were divided into six groups according to the interval from recognition of sepsis and septic shock to bundle achievement: group 1 (≤1 h; reference).” The SSC 1 hour bundle defines time zero as time of ED triage, the validity of which has been debated. Why are you defining “time zero” as time of “recognition of sepsis?” Please explain.
Response) Thank you for reviewer’s valuable comment. The time zero of the SSC 1-hr bundle was an ED triage when it was published in intensive care medicine in 2018. However, the time zero was changed to sepsis and septic shock recognition in October 2019. Probably, there are opinions that when the time zero is the ED triage, it is too difficult to achieve all five bundles in 1-hr in a practical clinical setting. We would like to attach the following website for reference (the SSC website does not currently use the term “time zero”).
https://www.sccm.org/SurvivingSepsisCampaign/Guidelines/Adult-Patients
The hour-1 bundle should be viewed as a quality improvement opportunity moving toward an ideal state. For critically ill patients with sepsis or septic shock, time is of the essence. Although the starting time for the Hour-1 bundle is recognition of sepsis, both sepsis and septic shock should be viewed as medical emergencies requiring rapid diagnosis and immediate intervention. ​ ​
The hour-1 bundle encourages clinicians to act as quickly as possible to obtain blood cultures, administer broad spectrum antibiotics, start appropriate fluid resuscitation, measure lactate, and begin vasopressors if clinically indicated. Ideally these interventions would all begin in the first hour from sepsis recognition but may not necessarily be completed in the first hour. Minimizing the time to treatment acknowledges the urgency that exists for patients with sepsis and septic shock. ​
↓↓↓
Also, “We defined the recognition of septic shock as the time of refractory hypotension (persistent hypotension despite administration of fluid challenge).” Please describe further. If someone presents to ED triage @ 13:00 and has SBP of 80mmHg and after 30cc/kg @ 13:45 they still have SBP of 82mmHg, was the time of refractory hypotension 13:00 or 13:45? Also, it seems problematic to define the time of starting the 1 hour bundle dependent on the completion of an element (fluid of the bundle). Isn’t this falsely inflating the length of the first hour and altering the analysis?
Response) The time of refractory hypotension is 13:45. This is because we defined refractory hypotension as persistent hypotension despite fluid administration. We agreed with reviewer’s concern that starting the 1-hr bundle as the timing of refractory hypotension seems problematic. Although the SSC guideline recommends starting bundle within 1-hr from the time of recognition of sepsis and septic shock, the criterion for recognition of sepsis was not clearly stated in SSC. Alternatively, if sepsis is defined when the SOFA score is acutely elevated by 2 points or more in patients with suspected or confirmed infection as reported in the sepsis 3 definition, the blood test results will require and take more time until recognition time. Hence, we defined it as the time of refractory hypotension recognition.
Results:
“A significantly lower proportion of in-hospital mortality was observed in the group that achieved a 1-hour bundle than the group that did not to achieve the bundle (13.8% vs. 19.9%, p = 0.004).” In the abstract, you state that there was no difference in mortality between the completed 1 hour bundle and didn’t complete 1 hour bundle groups. Which is it? I assume that these data are the univariate analysis. After reading multiple times, it is clear that the abstract is presenting the multivariable analysis results while this is presenting the univariate. I would present the Patient Characteristics, then the Univariate Analysis, then the Multivariable Logistic Regression and present clearly. Please clarify.
Response) Thanks for careful reading and sorry for reviewer’s inconvenient we revised table 1 and results section according to reviewer’s comment. Comparison of baseline characteristics and outcomes is presented in table 1. In-hospital mortality was not significantly different between 3 groups (13.8% vs. 16.9% vs. 20.1%, p = 0.075)
“In-hospital mortality was not significantly different between 3 groups (13.8% vs. 16.9% vs. 20.1%, p = 0.075)”
“The odds ratios (ORs) of 2-, 3-, 4-, 5-, and 6-hour bundle achievements for in-hospital mortality were significantly low (OR = 0.64, 0.603, 0.574, 0.569, and 0.511, respectively; p = 0.001, <0.001, <0.001, <0.001, and <0.001, respectively).” Compared to what as reference? Please clarify. This only becomes clear in the multivariable logistic regression analysis section where you let us know that you are comparing < 2 hour to > 2 hour; < 3 hour to > 3 hour, etc. Please correct so reads more clearly.
Response) Thanks for reviewer’s comments. This analysis is a univariate analysis that predicts in-hospital mortality by dividing the entire cohort into two groups (Table 2) (<2 vs. >2 hr, <3 vs. >3 hr and <4 vs. >4 hr, and so on, 2-group comparison). There is no specific reference. As reviewers judged that the results of this study were complicated and confusing, the comparison between the two groups presented the results of whether the achievement of 1, 3, and 6-hr bundle is valuable as a prognostic factor. We have revised this issue at results section as follows.
“The odds ratios (ORs) of 3- and 6-hr bundle achievements for in-hospital mortality were significantly low (OR = 0.603 and 0.511, respectively; p <0.001 both).
What are “others” that are comorbidities in Table 1 and used in the Multivariable Analysis? Please define others.
Response) Thanks for your review and comments. “others’ includes soft tissue, central nervous system, catheter related, blood stream and endocarditis. We have added the explanation about “others” in Table 1 as follows
“*Others includes soft tissue, central nervous system, catheter related, blood stream and endocarditis as infection site”
pp 7-8, multivariable analysis comparing 1 hour to the other time intervals for in-house, 28 day, and 90 day mortality. I find this very confusing. This relates to my concerns above about what is the comparison group. You haven’t articulated this clearly in Methods, that you are comparing <1 hour to > 1 hour, then < 2 hour to > 2 hour but also comparing < 1 hour as reference to > 1 hour, < 1 hour to > 2 hour, etc. If this is the approach you want to take, need to make much clearer. Also, please explain why (rationale for) you would do both?
Response) We are very sorry for the inconvenience. The purpose of our study was to determine if the achievement of the 1-hr bundle associated with a difference in prognosis than 2 hours or later. Initially, it was divided into 6 groups for <1, 1-2, 2-3, 3-4, 4-5 and 5-6 hours, and each delay of 1 hour was to determine how it affected the patient's prognosis (like previous study addressing prognostic value of antibiotics administration timing). However, since readers were confused in reading this study, we revised paper only about prognostic value of 1-, 3- and 6-hr bundles.
Pp 8-9: “Patients in whom blood culture was conducted within 1 and 2 h did not show a significant difference in in-hospital mortality compared to those for whom it was not (p = 0.228 and 0.087, respectively). Patients who underwent blood culture within 3, 4, 5, and 6 h had significantly lower in-hospital mortality rates than those who did not (p = 0.003, 0.007, 0.016, and 0.005, respectively).” Is this univariate or multivariable analysis? Also, again, compared to what as reference? Please clarify.
Response) This part is the result of individual element to predict outcomes using multivariable analysis. There is no reference because it is a comparison between two groups in the entire cohort (ex. blood culture <3 vs. >3-hr). The reference is a group that achieves the bundle within 1-hr, and is used to analyze whether the patient's outcome is different as the bundle achievement time is delayed.
However, since readers were confused in reading this study, we revised paper only about prognostic value of 1-, 3- and 6-hr bundles. Hence, we deleted all individual elements related information in results section.
Lactate within 5 hours, lower mortality. I find all of these time points for different variables very confusing and distracting from the central message of the manuscript. It feels like data dredging. If you explore enough variables, you will find significance. For these 1 hour bundle variables, you are exploring enough that you have to assume you will find something significant (1 in 20). I recommend narrowing down to the key variables you want to analyze.
Response) Thank you for reviewer’s valuable comments. According to reviewer’s suggestion we focused how the only 1-hr bundle achievement (<1 vs. >1 hr, 2 group comparison) associates with the patient's outcome. Patients were divided into 3 groups according to the time of bundle achievement and outcomes were compared to examine the difference in outcome over time: group 1 (≤1 h; reference), 2 (1–3 h) and 3 (3–6 h). Analysis of other cutoff (2-, 4- and 5-hr) were deleted. Please check the whole manuscript again.
Discussion:
“In a single-centre retrospective study, the group that failed to achieve severe sepsis and septic shock performance measure bundle (SEP-1) had a higher crude mortality than the group that achieved well, but no significant difference was found after adjusting for clinical variables and disease severity [20].” This statement mischaracterizes the Rhee study. It is not a single-centre study. It is a multicenter retrospective cohort study. Rhee, C.; Filbin, M.; Massaro, A.F.; Bulger, A.; McEachern, D.; Tobin, K.A.; Kitch, B.; Thurlo-Walsh, B.; Kadar, A.; Koffman, A. 
Compliance with the national SEP-1 quality measure and association with sepsis outcomes: a multicenter retrospective cohort study. Crit. Care Med. 2018, 46, 1585. Please correct.
Response) Thanks to your comments and we are sorry for the inconvenience. Reviewer’s point is correct. We have corrected the error as follows.
“In a multicenter retrospective cohort study, the group that failed to achieve severe sepsis and septic shock performance measure bundle (SEP-1) had a higher crude mortality than the group that achieved well, but no significant difference was found after adjusting for clinical variables and disease severity”
Evans study. Please justify the validity of comparing your adult study to the results of a pediatric sepsis study. How can you compare these results? I would recommend deleting reference to the Evans’ study.
Response) Evans’ study is a similar design that compared the bundle components within 1-hr. But we agreed with reviewer’s comment about pediatric population study Evans’s study has been deleted from the discussion section.
Limitations. There is no way to know that low compliance with the 1 hour bundle is because clinicians were unaware of the study. This may be the best compliance you can get due to limitations in meeting aspects of the bundle within 1 hour. I would delete this sentence.
Response) Thanks for reviewer’s suggestion. This sentence have been deleted.
“Third, it is unclear whether there was a start of rapid administration of 30 mL/kg crystalloid fluid within 1 hour of shock recognition because we defined the time of shock recognition as the time of refractory hypotension despite fluid administration. It could be argued that our analysis did not consists of five bundle treatments and that it consists of only four bundle treatment. However, it is likely that start of rapid fluid administration might be achieved within 1 h because the median time from ED triage to fluid completion was 87 min.” From my perspective, this is the biggest limitation of this study and its approach to data analysis. I derive a different conclusion from the definition of “the time of shock recognition as the time of refractory hypotension despite fluid administration.” From my perspective this means that your one hour window, as defined by SSC, is probably more likely a 2 hour window, etc. This needs to be clearly stated and I am not sure that you investigate a 1 hour window at all. Please comment and justify how you have actually addressed a 1 hour window.
Response) I totally agree with reviewer’s concerns. In our study, there is no certainty that rapid infusion was initiated within 1-hr due to limitations in retrospective data analysis. If time zero is the ED triage time, as you pointed out, this data will be verifying the prognostic value of 2-hr bundle achievement instead of 1-hr. However, in October 2019, time zero was changed to sepsis and septic shock recognition, and it is very difficult to recognize sepsis or septic shock from the time of ED triage in the clinical field. These data suggest that further investigation into the clinical implications of 1-hr bundle achievement in patients with septic shock is warranted.

Reviewer 2 Report
The authors studied the impact of 1-hour bundles achievement on outcomes in septic shock. This analysis follows on from the 2018 supplement of the SSC and remains highly discussed at present both for questions of scientific evidence but also for feasibility and implementation in emergency deparment. This study including 1,612 patients with septic shock provides additional information in the field. Although of interest, it raises the following comments:
General comments:
- Why have separated the cohort into 6 groups when the SSC recommendations were initially based on 6h then 3h-bundle? This confuses the paper and brings in irrelevant data. It is important that the authors simplify the presentation of the results by comparing only 3 groups as validated in the literature (<1h: test group, <3h: 1st control group and <6h: 2nd control group)
- If we want to have a clear and precise message for the readers, we must greatly simplify the results and keep only the relevant data and which allow to have a precise opinion on the analysis. There is a lot of data which ends up making the paper very statistical but not very clinical and practical. It is important that authors clarify this point.
Specific comments:
Abstract: ok
Introduction:
This section is well-written
Methods:
Study design: ok
Setting: This paragraph can be merged with the paragraph "study design" in order to facilitate reading
Definitions and outcomes: ok
Statistical analysis: See point #1
Results:
There are too many results making it difficult to read the study. To be adapted according to advice number 1
Merge table 2 and 3 into one single table allowing to have the univariate and the multivariate analysis concerning the bundles. Change the global univariate analysis to supplemental data
Table 1: See point #1
Table S1: Irrelevant data
Table S2: See point #1
Table S3: irrelevant data
Table S4: irrelevant data
Discussion:
To be adapted according to the new results and the new presentation.
The third point of the limits deserves to be better explained. The last point has little interest
Conclusion:
The conclusion is relevant and well-written but could be adapted among the modifications.
Minor comments:
Nothing
Author Response
Reviewer 2
General comments:
- Why have separated the cohort into 6 groups when the SSC recommendations were initially based on 6h then 3h-bundle? This confuses the paper and brings in irrelevant data. It is important that the authors simplify the presentation of the results by comparing only 3 groups as validated in the literature (<1h: test group, <3h: 1st control group and <6h: 2nd control group)
Response) Thank you for your reviews and comments. I totally agree with your concern. Instead of dividing the cohort into six groups, we have re-analyzed comparing the outcome of 1-, 3- and 6-hr bundle, and revised the text and table, so please check.
- If we want to have a clear and precise message for the readers, we must greatly simplify the results and keep only the relevant data and which allow to have a precise opinion on the analysis. There is a lot of data which ends up making the paper very statistical but not very clinical and practical. It is important that authors clarify this point.
Response) We are very sorry for the inconvenience. According to reviewer’s recommendation, we analyzed again how the only 1-hr bundle achievement (<1 vs. >1-hr, 2 group comparison in whole cohort) associates with the patient's outcome. Patients were also divided into 3 groups according to the time of bundle achievement and outcomes were compared to examine the difference in outcome over time: group 1 (≤1 hour; reference), 2 (1–3 hour) and 3 (3–6 hour). Other analysis parts were deleted. Please check the whole manuscript and figure again.
Specific comments:
Abstract: ok
Introduction:
This section is well-written
Methods:
Study design: ok
Setting: This paragraph can be merged with the paragraph "study design" in order to facilitate reading
Response) Thanks again for reviewer’s valuable comments. As recommended, the setting part was combined into the study design part.
Results:
There are too many results making it difficult to read the study. To be adapted according to advice number 1
Response) The analysis method and volume of the paper have been simplified according to your recommendations.
Merge table 2 and 3 into one single table allowing to have the univariate and the multivariate analysis concerning the bundles. Change the global univariate analysis to supplemental data
Response) As your recommendation, tables 2 and 3 are combined into a single table. We changed the global univariate analysis to supplemental data.
Table 1: See point #1
Table S1: Irrelevant data
Table S2: See point #1
Table S3: irrelevant data
Table S4: irrelevant data
Response) Following your recommendation, We deleted the table S1, S3 and S4.
Discussion:
To be adapted according to the new results and the new presentation.
Response) We have revised a discussion section according to the new research and the new presentation. Please check.
The third point of the limits deserves to be better explained. The last point has little interest
Response) I agree with your concern that starting the 1-hr bundle as the timing of refractory hypotension seems problematic. Although the SSC guideline recommends starting bundle within 1-hr from the time of recognition of sepsis and septic shock, the criterion for recognition of sepsis was not clearly stated in SSC. Alternatively, if sepsis is defined when the SOFA score is acutely elevated by 2 points or more in patients with suspected or confirmed infection as reported in the sepsis 3 definition, the blood test results will require and take more time until recognition time. Hence, we defined it as the time of refractory hypotension recognition. According to reviewer’s suggestion, the last sentence of limitation was deleted. We revised this issue at limitation section as follows.
“In the SSC guidelines, it is mandated that rapid fluid administration should be initiated within 1-hr of shock recognition. We believe that start of fluid administration was performed as rapid as fast given median time”
Conclusion:
The conclusion is relevant and well-written but could be adapted among the modifications.
Minor comments:
Nothing

Round 2
Reviewer 1 Report
Thank you for a careful revision of the manuscript.
You have addressed my major concerns and improved the manuscript significantly.
Reviewer 2 Report
The authors responded well to comments.